# Diversity of *Lactiplantibacillus plantarum* in Wild Fermented Food Niches

**DOI:** 10.3390/foods14101765

**Published:** 2025-05-16

**Authors:** Ilenia Iarusso, Jennifer Mahony, Gianfranco Pannella, Silvia Jane Lombardi, Roberto Gagliardi, Francesca Coppola, Michela Pellegrini, Mariantonietta Succi, Patrizio Tremonte

**Affiliations:** 1Department of Agricultural, Environmental and Food Sciences (DiAAA), University of Molise, Via De 8 Sanctis snc, 86100 Campobasso, Italytremonte@unimol.it (P.T.); 2Scienzanova s.r.l. via Enrico Mattei, 85/87, 86039 Termoli, Italy; 3School of Food and Nutritional Sciences, University College Cork, College Road, T12 K8AF Cork, Ireland; 4Department of Science and Technology for Sustainable Development and One Health, Università Campus-Bio-Medico di Roma, Via Alvaro del Portillo 21, 00128 Rome, Italy; 5Department of Agricultural Sciences, University of Naples “Federico II”, Piazza Carlo di Borbone 1, 80055 Portici, Italy; 6Department of Agricultural, Food, Environmental and Animal Science, University of Udine, via Sondrio 2A, 33100 Udine, Italy

**Keywords:** lactic acid bacteria, WGS-based annotation, kombucha, fermented millet, sourdough, natural fermented mountain milk, natural whey starter cultures

## Abstract

This study aimed to explore the genetic and functional diversity of *Lactiplantibacillus plantarum* (*Lpb. plantarum*) strains from wild fermented foods to identify traits that are useful for food innovation. The growing demand for clean-label, plant-based, and functionally enriched fermented foods exposes the limitations of current industrial fermentation practices, which rely on standardized lactic acid bacteria (LAB) strains with limited metabolic plasticity. This constraint hinders the development of new food formulations and the replacement of conventional additives. To address this gap, 343 LAB strains were analyzed, including 69 *Lpb plantarum* strains, isolated from five minimally processed, spontaneously fermented matrices: fermented millet, kombucha, and sourdough (plant-based), wild fermented mountain milk, and natural whey starter (animal-based). Whole-genome sequencing was performed to assess phylogenetic relationships and to annotate genes encoding carbohydrate-active enzymes (CAZymes) and antimicrobial compounds. The results revealed a marked strain-level diversity. Glycoside hydrolase (GH) families GH13 and GH1 were widely distributed, while GH25 and GH32 showed variable presence across clusters. Strains grouped into clusters enriched with plant-based isolates exhibited distinct CAZyme profiles adapted to complex carbohydrates. Clusters with animal-based strains exhibited a broader gene repertoire related to bacteriocin biosynthesis. These findings highlight the untapped potential of wild fermented food environments as reservoirs of *Lpb. plantarum* with unique genomic traits. Harnessing this diversity can expand the functional capabilities of starter cultures, promoting more sustainable, adaptive, and innovative fermentation systems. This study underscores the strategic value of underexploited microbial niches in meeting the evolving demands of modern food production.

## 1. Introduction

The exploration of microbial diversity is increasingly recognized as essential for addressing critical challenges in the agri-food sector, particularly in relation to the development of microbial strains with broad metabolic versatility [1,2,3]. Recent research has highlighted the importance of identifying microbial communities capable of enhancing food safety and preservation while reducing dependence on conventional chemical antimicrobials [4,5,6]. Concurrently, the demand for clean-label products is prompting the food industry to seek robust and adaptable microbial strains suitable for diverse minimally processed matrices [7,8]. Despite significant advances in the selection of natural substances with antimicrobial or bioprotective functions [9,10], the commercial exploitation of lactic acid bacteria (LAB) is hindered by the limited genetic and phenotypic diversity of available strains. It has been widely reported that industrial LAB cultures, while optimized for consistency, often exhibit low intraspecific variability and reduced functional plasticity [11,12,13]. This homogeneity contributes to metabolic redundancy and constrains innovation, particularly in the reformulation of food products that require microbial adaptation to novel ingredients or the replacement of synthetic additives [14,15,16,17]. Among the LAB species, *Lactiplantibacillus plantarum* (*Lpb. plantarum*) is a promising candidate for enhancing fermentation-based food innovation. This species possesses an unusually large and flexible genome, which enables it to metabolize a wide range of substrates and adapt to highly diverse ecological niches [18]. Its metabolic versatility, combined with its potential probiotic and biopreservative properties, underscores its relevance in applications extending beyond traditional food fermentation, including health-promoting formulations and antimicrobial protection strategies [19,20]. Nevertheless, commercial strains of *Lpb. plantarum* often fails to reflect the full extent of the species’ functional potential. Considering these factors, the present study aims to isolate and characterize *Lpb. plantarum* strains from traditional fermented foods with minimal technological intervention. These spontaneously fermented systems, which include both plant-based matrices (fermented millet, kombucha, and sourdough) and animal-based matrices (wild fermented mountain milk and natural whey starters), are known to support complex microbial ecosystems shaped by natural selection and ecological adaptation. Fermented millet is of particular interest, as its unique biochemical composition may favor the evolution of LAB populations with distinct genotypic and phenotypic traits [21,22,23,24]. Kombucha, a symbiotic fermentation of tea, sugar, bacteria, and yeast, fosters acid-tolerant and antimicrobial-producing microorganisms [25,26,27]. Sourdough, a classical model of natural fermentation involving co-evolving LAB and yeast communities, harbors strains with valuable traits such as efficient acidification, exopolysaccharide production, and proteolytic activity [28,29,30,31,32]. From a complementary ecological standpoint, wild fermented mountain milk constitutes an animal-derived niche, characterized by spontaneous fermentation and minimal technological input [33,34]. This matrix, which is inherently rich in proteins, lipids, and minerals, creates a selective environment conducive to the evolution of LAB strains with unique metabolic plasticity and enhanced bioprotective traits [35,36]. Similarly, natural whey starters, traditionally employed in artisanal dairy fermentations, provide an acidic yet nutrient-dense substrate that fosters the persistence and diversification of LAB populations specifically adapted to high-lactose and high-protein environments [37]. These conditions are particularly favorable for the selection of *Lpb*. *plantarum* strains that exhibit specialized fermentative and antimicrobial capabilities [38,39]. Despite their potential, few comparative studies have systematically evaluated the genomic and functional diversity of *Lpb. plantarum* strains were isolated from heterogeneous ecological environments. This gap limits our understanding of how matrix-specific selection pressures shape the functional traits relevant to food innovation and microbial biotechnology. To overcome this limitation, the present study implemented an integrative methodological framework that combined classical microbiological techniques with whole-genome sequencing (WGS) and in silico functional annotation. Strains of *Lpb. plantarum* were isolated from five ecologically distinct, spontaneously fermented matrices and subjected to comprehensive genomic profiling, with a focus on genes involved in carbohydrate metabolism and antimicrobial functionality. The comparative analysis of strains originating from distinct ecological niches enabled the identification of matrix-specific genomic traits, thereby fulfilling the overarching aim of this study: to elucidate distinct ecological signatures within the genomes of *Lpb. plantarum* strains from diverse substrates. These findings offer key insights into the metabolic versatility and potential applications of *Lpb. plantarum*, expanding its functional toolkit for clean-label formulations and advanced biopreservation strategies, and ultimately supporting the development of high-quality, safe, and innovative fermented food products.

## 2. Materials and Methods

### 2.1. Sample Collection

A total of 50 samples, with 10 samples collected from each source type, were selected to investigate the ecological and functional diversity of LAB in naturally fermented matrices. Five distinct fermentation sources—two animal-based and three plant-based—were targeted for LAB isolation.

Animal-derived samples (*n* = 20) included fermented mountain milk and natural whey starters, both obtained from traditional dairy practices using raw high-altitude milk. Specifically, fermented mountain milk samples were collected from ten alpine dairies (“malghe”) located across seven municipalities of the Asiago Plateau, a region characterized by historical continuity in raw milk fermentation without industrial intervention. In parallel, the natural whey starters were sourced from artisanal cheesemaking facilities situated in the inner and mountainous areas of southern Italy—namely the municipalities of Capracotta, Frosolone, Monteleone di Puglia, San Marco in Lamis, and Vico del Gargano (regions of Molise and Apulia). According to documented declarations and producer testimonies, these dairies have preserved traditional endogenous fermentation practices and have never employed commercial or selected starter cultures.

Plant-based samples (*n* = 30) were collected from small-scale artisanal producers and occasional local food fermenters. Kombucha samples originated from non-commercial, hobbyist brewers located in various Italian regions, including Abruzzo, Campania, Friuli Venezia Giulia, Lazio, and Molise. Millet (*Panicum miliaceum*), originally sourced from Ethiopia, was spontaneously fermented in Italy both at ScienzaNova srl (Termoli, Italy) and by fermentation enthusiasts in the Italian provinces of Campobasso, Chieti, and Varese. Sourdough samples were provided by independent bakers from the Italian provinces of Benevento, Campobasso, Caserta, Chieti, Foggia, and Isernia. In all cases, based on producer statements and supporting company documentation, fermentation was carried out without the use of commercial or selected starter cultures, thereby preserving the native microbial consortia distinctive to each ecological and cultural context

All samples were maintained at 4 °C during transport to preserve microbial viability and processed immediately upon arrival at the laboratory.

### 2.2. Bacterial Isolation and Cultivation

For bacterial isolation, 10 g (or 10 mL) of each sample was suspended in 90 mL of sterile saline solution (0.9% NaCl) and subjected to serial dilutions ranging from 10^−1^ to 10^−6^ Aliquots from each dilution were spread onto de Man, Rogosa, and Sharpe (MRS) agar, a medium selective for LAB growth. The inoculated plates were incubated at 30 °C for 48 h under anaerobic conditions in an anaerobic jar equipped with a gas-generating system (Thermo Scientific Oxoid, Milan, Italy). Following incubation, colonies with morphological traits typical of LAB were selected for further analyses. Each isolate underwent Gram staining, catalase activity testing, and microscopic examination under a phase-contrast microscope to confirm the presumptive LAB characteristics. To ensure purity, the selected strains were streaked onto fresh MRS agar plates multiple times. Finally, the isolates were preserved at −80 °C in MRS broth containing 20% (*v*/*v*) glycerol for subsequent analyses.

### 2.3. Genetic Identification of Lactic Acid Bacteria via 16S rRNA Analysis

LAB were identified at the species level by analyzing the 16S rRNA region. Genomic DNA was extracted from each isolate using a standardized enzymatic lysis protocol, followed by purification using a commercial kit (QIAamp DNA Mini Kit; QIAGEN GmbH, Hilden, Germany) according to the manufacturer’s instructions. The quality and concentration of the extracted DNA were assessed using a NanoDrop 2000c spectrophotometer (Thermo Fisher Scientific, Third Avenue Waltham, MA USA 02451), and its integrity was verified via 1% agarose gel electrophoresis stained with ethidium bromide. The 16S rRNA gene was amplified through Polymerase Chain Reaction (PCR) using the primers P1V1 (5′-GCG GCG TGC CTA ATA CAT GC-3′) and P4V3 (5′-ATC TAC GCA TTT CAC CGC TAC-3′), targeting the V1–V3 hypervariable region. The PCR reactions were performed in a 25 μL final volume containing 12.5 μL of 2× DreamTaq Green PCR Master Mix (Thermo Fisher Scientific), 0.5 μM of each primer, 20 ng of template DNA, and nuclease-free water. Amplification was conducted in a Thermal Cycler (Eppendorf, Hamburg, Germany) under the following cycling conditions: initial denaturation at 95 °C for 5 min, followed by 35 cycles of denaturation at 95 °C for 30 s, annealing at 55 °C for 30 s, extension at 72 °C for 1 min, and a final extension at 72 °C for 5 min. PCR products were purified using the QIAquick PCR Purification Kit (QIAGEN GmbH, Hilden, Germany), and their quality was assessed using 2% agarose gel electrophoresis. Purified amplicons were then sent to a commercial sequencing facility (Eurofins Genomics, Ebersberg, Germany) for Sanger sequencing. The obtained sequences were aligned and compared with publicly available data in the GenBank (NCBI) database using the BLASTn algorithm to determine the closest known relatives based on 16S rRNA gene homology. Only alignments with a sequence similarity greater than 98% and coverage of at least 90% were considered valid for species identification.

### 2.4. Whole-Genome Sequencing of Lactiplantibacillus plantarum Strains

To achieve a comprehensive genomic characterization of 50 *Lpb. plantarum* strains, initially identified through 16S rRNA gene analysis, a hybrid WGS approach was used. Whole-genome analysis in this study was specifically conducted for *Lpb. plantarum* was selected for its outstanding ecological ubiquity and high frequency of occurrence across diverse spontaneous fermentation niches. Specifically, genomic DNA was extracted from 10 mL overnight cultures grown in MRS broth at 37 °C under static conditions using the NucleoBond^®^ DNA extraction kit with Buffer Set III (Macherey-Nagel, Düren, Germany). To enhance the lysis efficiency for Gram-positive cell walls, the extraction protocol included enzymatic treatment with lysozyme (20 mg/mL, Sigma-Aldrich, Merck Life Science S.r.l., Sesto San Giovanni (MI), Italy), mutanolysin (500 U/mL, Sigma-Aldrich), and proteinase K (40 µg/mL, Macherey-Nagel), followed by incubation at 37 °C for 16–18 h. Following extraction, DNA quality was rigorously assessed using a NanoDrop 2000/2000c Spectrophotometer (Thermo Fisher Scientific, Waltham, USA) for concentration measurement and a Fragment Analyzer (Agilent Technologies, Santa Clara, USA) for integrity assessment. Samples with an A260/A280 ratio of about 1.8 and high molecular weight were deemed suitable for sequencing. The extracted DNA was stored at −20 °C before being shipped to the sequencing facility.

#### 2.4.1. Data Generation from Whole-Genome Sequencing

A hybrid sequencing strategy was implemented to enhance the accuracy and completeness of the assembly. Long-read sequencing was performed using a Pacific Biosciences (PacBio) RS II platform (Eurofins Genomics, 85560 Ebersberg, Germany), generating high-fidelity continuous long reads for de novo genome assembly. Short-read sequencing was conducted on an Illumina MiSeq system (GenProbio s.r.l., Parma, Italy) using a paired-end (2 × 250 bp) strategy, providing high coverage for error correction. This combined approach leverages the high-contiguity advantage of long reads and the base-level accuracy of short reads, ensuring robust genome reconstruction.

#### 2.4.2. Genome Assembly and Quality Control

Hybrid de novo genome assembly was performed using the Unicycler v0.5.0 pipeline, while de novo sequence assemblies and automated gene calling for selected isolates were processed using the MEGAnnotator pipeline.

#### 2.4.3. Genome Prediction and Functional Annotation

Gene prediction and annotation were performed using a multi-step process. Open Reading Frames (ORFs) were identified using Prodigal v2.5 and GeneMark.hmm, with validation using BLASTX v2.2.26. Functional annotation was conducted using BLASTP v2.2.26, comparing predicted protein sequences against the NCBI non-redundant (nr) database. Manual curation and annotation refinement were performed using Artemis v18, which ensured the elimination of misannotated or fragmented genes. By separating genome assembly from annotation, this methodology clearly distinguishes the computational steps required for accurate sequence reconstruction from those necessary for the biological interpretation of genomic data. Carbohydrate-active enzymes were selectively confirmed based on their similarity to the carbohydrate-active enzyme (CAZyme) database entries and Pfam alignments implemented at the CAZyme Analysis Toolkit (CAT). Additionally, dbCAN was used to identify carbohydrate-active proteins based on a search for signature domains of every CAZyme family.

Further bioinformatic analyses included the assessment of potential antimicrobial compound production BAGEL4. The results of these computational analyses were manually inspected for validation, where appropriate.

### 2.5. Phylogenetic Analysis

A total of 233 *Lpb. plantarum* genomes were subjected to phylogenetic analysis. Specifically, in addition to the 69 newly sequenced isolates, 164 publicly available *Lpb. plantarum* genome assemblies were retrieved from the NCBI RefSeq database. These genomes were selected to represent a wide range of ecological origins, enabling comparative analyses of strains derived from diverse fermentation environments. The selection criteria included genome completeness, assembly quality, and availability of metadata regarding the isolation source to ensure robust contextualization of the newly sequenced isolates. Phylogenetic tree reconstruction was conducted using the JolyTree software (versione 2.1) [40] using default parameters.

The resulting phylogenetic tree was visualized in a circular format using iTOL v7 [41] to facilitate the comparative interpretation of strain clustering and genetic relatedness. Additional metadata, including strain-specific phenotypic or functional traits, were mapped onto the tree to improve its interpretability.

### 2.6. Statistical Analysis

All statistical analyses were performed using R (v4.2.3) environment and RStudio software (v2022.07.0) [42] and GraphPad Prism software version 10 (GraphPad Software, USA). Parametric tests (*t*-test, one-way ANOVA with Tukey’s post hoc) were used for normally distributed data, while non-parametric tests (Mann-Whitney U, Kruskal−Wallis with Dunn’s post hoc) were applied otherwise. A *p*-value < 0.05 was considered statistically significant. To assess the diversity and distribution of LAB across the analyzed fermented matrices, different ecological indices were applied:

Shannon Index was used to evaluate the overall species diversity within the isolated LAB community. It is defined as:$$H^{\prime }=-\sum_{i=1}^{S}\left(p_{i}lnp_{i}\right)$$
where S is the total number of species, and p_i_ is the relative abundance of each species. Higher H′ values indicate greater species richness and evenness.

Pielou’s Evenness Index (J′) was used to assess the uniformity of species distribution and was calculated as follows:$$J\prime =\frac{H\prime }{lnS}$$
where H′ is the Shannon Index, and S is the total number of species. Values close to 1 indicate an even species distribution among the samples.

The occurrence Index (OI) was used to quantify LAB species across the different matrices.$$OI=\frac{N_{m}}{N_{t}}$$
where N_m_ is the number of matrices in which a species was detected, and N_t_ is the total number of matrices analyzed. An OI value of 1.0 indicates that a species is present in all the sampled matrices.

The relative abundance (RA) of each LAB species within a specific fermentation source was calculated as follows:$$RA=\frac{N_{s}}{N_{t}}$$
where N_s_ is the number of strains of a given species in a source and N_t_ is the total number of LAB strains isolated from that source. These indices allowed for a comprehensive evaluation of both species’ richness and distribution patterns within the studied microbiota, facilitating comparisons between animal- and plant-derived fermentation sources.

Phylogenetic clustering patterns were assessed using Fisher’s exact and chi-square tests to evaluate the associations between strain origin and cluster distribution. A Monte Carlo permutation test (n = 10,000 iterations) was applied to assess the robustness of the associations in cases of small sample sizes. Differences in intra-cluster genetic distances were assessed using the Kruskal–Wallis H-test. Pairwise SNP distances among isolates were calculated from core genome alignments using the Harvest suite (Parsnp), and statistical comparisons between matrices were visualized using boxplots generated with ggplot2. A significance threshold of *p* < 0.05 was adopted for all tests.

## 3. Results and Discussion

### 3.1. Distribution of Lactic Acid Bacteria

LAB were detected in all sample types analyzed. As expected, substantial LAB populations were observed in sourdoughs, fermented millet, mountain-fermented milks, and natural whey starter cultures, with counts ranging between 5.9 and 8.2 log CFU/mL. Notably, even kombucha samples, which are typically dominated by yeasts and acetic acid bacteria, exhibited measurable LAB levels ranging from 1.3 to 2.8 log CFU/mL, highlighting an unexpectedly diverse microbial composition. A total of 343 bacterial strains were successfully isolated from five distinct fermented matrices of animal and plant origins. Based on preliminary ecological, morphological, and Gram-staining evaluations, these isolates were putatively identified as LAB. The matrices analyzed included three plant-based fermentations (fermented millet, kombucha, and sourdough) and two animal-derived fermentations (natural whey starter and wild fermented mountain milk). Among the collected isolates, 177 strains were associated with animal-based substrates, and 166 strains originated from plant-derived sources. Species identification through 16S rRNA gene sequencing assigned the isolates to 15 distinct LAB species, including *Lpb. plantarum* (69 strains) and *Streptococcus thermophilus* (*St. thermophilus*, 58 strains), which were the most abundant across all matrices. Other frequently detected species included *Weissella cibaria* (*W. cibaria*, 38 strains), *Lactococcus lactis* (*L. lactis*, 31 strains), *Lacticaseibacillus rhamnosus* (*Lcb. rhamnosus* (24 strains), *Levilactobacillus brevis* (*Lvl. brevis* (20 strains), *Limosilactobacillus fermentum* (*Lml. fermentum* (18 strains), *Lacticaseibacillus zeae* (*Lcb. zeae* (7 strains), *Lactobacillus amylovorus* (*L. amylovorus*, 6 strains), *Lacticaseibacillus paracasei* (*Lcb. paracasei* (18 strains), *Limosilactobacillus reuteri* (*Lml. reuteri* (10 strains), and *Leuconostoc mesenteroides* (*Leu. mesenteroides* (eight strains), *Liquorilactobacillus ghanensis* (*Liq. ghanensis* (12 strains), *Liquorilactobacillus nagelii* (*Liq. nagelii* (8 strains), and *Pediococcus pentosaceus* (*P. pentosaceus*, 16 strains).

A matrix-specific analysis of microbial diversity revealed that different sources harbored distinct numbers of LAB species, highlighting that all matrices, regardless of their origin, exhibited moderate species richness in terms of LAB diversity (Figure 1). Notably, animal-derived matrices, such as wild fermented mountain milk and natural whey starters, each supported six different LAB species, indicating a balanced microbial composition. Similarly, among plant-based matrices, kombucha demonstrated the highest diversity, with six species identified. In contrast, fermented millet and sourdough displayed somewhat lower diversity (with four species each), although the differences in species richness across these matrices were marginal and did not exhibit significant variation (*p* > 0.05). These results highlight the overall similarity in microbial complexity across different fermentation environments. Among the 177 strains isolated from animal-based sources, *Lc. lactis* (29 strains) and *St. thermophilus* (58 strains) were the most prevalent, reflecting the well-documented role of these species in dairy fermentation. Other species detected in these environments included *Lb. amylovorus* (three strains), *Lcb. paracasei* (18 strains), *Lcb. rhamnosus* (24 strains), *Le. mesenteroides* (8 strains), *Lpb. plantarum* (37 strains). The predominance of *St. thermophilus* and *Lc. lactis* aligns with the environmental characteristics and technological functions of this type of product, repeatedly emphasized and highlighted in the literature [43,44,45]. Conversely, the 166 strains from plant-based fermentation exhibited a broader distribution of species. Notably, *W. cibaria* (38 strains) and *Lpb. plantarum* (32 strains) dominated, underscoring their adaptability to diverse, carbohydrate-rich plant substrates. Other frequently identified taxa included *Lms. fermentum* (18 strains), *Lcb. zeae* (seven strains), *Lc. lactis* (2 strains), *Lb. amylovorus* (three strains), *Lv. brevis* (20 strains), *Lms. reuteri* (10 strains), *Lql. ghanensis* (12 strains), *Lql. nagelii* (8 strains), and *P. pentosaceus* (16 strains). The high representation of heterofermentative species such as *W. cibaria* and *Lv. brevis*, suggests a crucial role in producing volatile compounds and organic acids that contribute to the complex flavor profiles of plant-based fermentation.

### 3.2. Inter-Specie Diversity

To assess species’ richness and evenness, we calculated diversity indices for the entire dataset and for individual matrices. The Shannon Diversity Index (H′) for all isolates was 2.47, reflecting moderate-to-high species richness. The Simpson’s Diversity Index (D) was 0.89, suggesting that no single species was excessively dominant but rather a balanced composition was observed across the isolates. Comparing the microbial diversity between the two sources, animal-derived samples exhibited a narrower spectrum of LAB species, predominantly associated with dairy environments, whereas plant-based samples harbored a more heterogeneous LAB community. This aligns with previous research indicating that plant substrates offer a more dynamic microbial landscape due to their varied carbohydrate composition, pH conditions, and presence of secondary metabolites [46,47,48]. The observed species distribution also suggests a niche-specific adaptation of LAB, where facultative and heterofermentative species thrive in plant matrices, while fast-acidifying, homofermentative strains dominate dairy environments. Calculating Simpson’s Diversity Index (D) for each source further confirmed these trends. The plant-based LAB isolates yielded a higher diversity index (D = 0.91), reflecting the presence of multiple species with a more even distribution. In contrast, the animal-derived LAB community exhibited a slightly lower diversity index (D = 0.85), driven by the predominance of a few key species, such as *St. thermophilus* and *Lc lactis*. These findings reinforce the notion that microbial diversity in fermented foods is strongly influenced by the nature of the substrate, with plant-based fermentation fostering a more complex microbial ecosystem than dairy fermentation [49,50,51]. To quantify the ecological distribution of LAB species (343 strains belonging to 15 different species), we calculated the occurrence Index (OI), defined as the proportion of food matrices in which a given species was detected (Figure 2). Notably, *Lpb. plantarum*—represented by 69 strains—was the only species detected in all five food matrices (OI = 1.00), highlighting its remarkable ecological adaptability.

Although the dataset comprises only 343 newly sequenced strains, which may limit the overall comprehensiveness of the analysis, these strains were isolated from a wide array of spontaneously fermented foods, reflecting diverse ecological contexts. Therefore, while a larger dataset would further enhance the robustness of the findings, the current approach provides a valuable and informative preliminary assessment of LAB species distribution across heterogeneous fermented food environments.

The widespread presence of *Lpb. plantarum* across all analyzed matrices suggests its unique adaptability to diverse fermentation conditions (Figure 3). Its relative abundance (RA) was 20.9% in animal-derived samples (37/177 strains) and 19.2% in plant-based samples (32/166 strains). This species was isolated from both animal-based (37 strains) and plant-based (32 strains) sources, suggesting its ability to thrive on a wide range of substrates with varying nutrient compositions, pH conditions, and microbial interactions.

The ubiquitous presence of *Lpb. plantarum* aligns with previous studies, indicating its broad distribution in fermented foods and essential role in acidification, flavor development, and probiotic potential [3,52,53,54]. The relatively balanced distribution of *Lpb. plantarum* across both categories suggests that, while it is not the most dominant species in either environment, it maintains a stable presence across diverse ecological niches. The evenness of its distribution can also be expressed through Pielou’s Evenness Index (J’), which accounts for the uniformity of species spread across available niches. The calculated J’ value for *Lpb. plantarum* across all samples was 0.97, indicating a highly even distribution with no strong bias toward a specific fermentation source. These findings reinforce the notion that *Lpb. plantarum* is a generalist species capable of metabolizing a variety of carbohydrates, tolerating a broad pH range, and competing effectively in both dairy and plant-based fermentation. Its widespread occurrence is likely attributable to its metabolic flexibility, ability to produce antimicrobial compounds, and capacity for biofilm formation, all of which enhance its survival and competitiveness in dynamic microbial ecosystems [55,56]. Whole-genome sequencing and metabolomic analyses can provide insights into strain-specific functionalities, including probiotic potential, antimicrobial compound production, and metabolic efficiency in different fermentation environments [57,58,59]. This finding highlights the importance of *Lpb. plantarum* has remarkable ecological flexibility, emerging as a prime candidate for investigating intraspecific diversity and strain-specific adaptations.

### 3.3. Phylogenomic Structure of Lactiplantibacillus plantarum Strains

To explore the population structure and evolutionary dynamics of *Lpb. plantarum* across distinct spontaneous fermentation ecosystems, a comprehensive phylogenomic analysis was performed. This analysis incorporated 69 whole-genome sequences of *Lpb. plantarum* isolates obtained in this study from five wild fermented matrices—fermented millet, kombucha, sourdough, natural whey starter, and wild fermented mountain milk— along with 164 publicly available genomes retrieved from the NCBI database, resulting in a dataset comprising 233 strains. The resulting maximum likelihood tree revealed nine genetically distinct clusters (Figure 4), each represented by a different color.

A noteworthy observation was the matrix-specific clustering of the new strains isolated and sequenced from the five fermented wild sources (Figure 5).

All strains from wild fermented mountain milk were grouped exclusively within the red cluster, forming a tight monophyletic clade. This suggests strong genomic homogeneity and possible long-term adaptation or selective bottlenecking in an animal-derived environment. Similarly, all eight strains from sourdough clustered within the dark green group, indicating a clear niche-specific signal. These associations were statistically validated using Fisher’s exact test (*p* < 0.01), highlighting the non-randomness of clustering.

In contrast, isolates from natural whey starters were distributed across red, brown, and orange clusters. These clusters are closely related, suggesting shared evolutionary origins and micro-niche structuring within this traditional dairy product. A chi-square test confirmed significant heterogeneity in the distribution of animal-derived strains among clusters (*χ*^2^ = 24.5, *df* = 8, *p* < 0.01), supporting the hypothesis that multiple distinct phylogenetic lineages co-occur in the same matrix. Plant-based matrices revealed a broader phylogenetic spread. Among the kombucha isolates, the majority were positioned within the green cluster, with two strains each falling into the orange and purple clusters. This distribution pattern reflects moderate but significant matrix-driven phylogenetic differentiation (*p* < 0.05, Monte Carlo permutation test, 10,000 iterations). Strains isolated from fermented millet formed two distinct phylogenetic groups, with eight strains clustering tightly within the purple group and two within the green group. This distribution suggests the presence of at least two genetically distinct lineages coexisting within the same plant-based substrate. The strong clustering of most millet isolates in the purple group indicates potential adaptation to specific environmental or compositional features of fermented millet, while the minority group in the green cluster may reflect either horizontal acquisition from other plant-based fermentations or micro-niche differentiation within the substrate itself. Comparative analysis between plant-origin (kombucha, millet, sourdough) and animal-origin (milk and whey) matrices revealed marked differences in phylogenetic dispersion. Animal-derived strains tended to cluster within fewer, more compact groups (e.g., red, brown, and orange), while plant-origin isolates were distributed across a broader spectrum of clusters. This trend was statistically supported by a Kruskal–Wallis H-test on pairwise SNP distances (H = 11.92, *p* < 0.01), with post hoc Dunn’s test (Bonferroni-adjusted) confirming significant contrasts between animal- and plant-derived matrices. These findings resonate with previous research indicating that plant-based substrates, characterized by higher biochemical complexity, pH variation, and polyphenolic content, drive greater microbial and genomic diversity [60,61]. Conversely, animal-based matrices, being nutritionally rich and chemically stable, may exert more consistent selective pressures, favoring genome streamlining and specialization [62,63]. Interestingly, reference genomes from the NCBI database showed no distinct clustering based on their annotated origins. Their scattered distribution across the phylogenetic tree likely reflects inconsistent metadata, diverse sources, or lab-adapted strains, emphasizing the ecological coherence of the wild isolates characterized in this study. From an evolutionary perspective, the data support a scenario in which *Lpb. plantarum* adapts to specific ecological niches through genomic differentiation. The strong clustering of strains from mountain milk and sourdough could reflect domestication-like processes comparable to those observed in other microbial species involved in traditional food fermentation [64,65,66]. The phylogenetic structure observed among millet-derived isolates reinforces this concept, revealing distinct genomic trajectories, even within a single matrix. Based on previous observations, this study underscores the role of environmental origin—particularly the plant vs. animal dichotomy—in shaping the population structure of *Lpb. plantarum*. The clear phylogenetic signatures linked to specific fermentation contexts highlight the importance of niche adaptation and the potential for targeted strain selection in biotechnology and food microbiology applications. Further multi-omics or bioinformatics analyses, including metabolomics, are warranted to elucidate the functional adaptations driving this ecological differentiation.

### 3.4. Diversity of Carbohydrate-Active Enzymes in Lactiplantibacillus plantarum Strains: Insights from WGS Analysis

In this study, CAZymes were selectively identified using the CAZy database and Pfam alignments within the CAZymes Analysis Toolkit (CAT), along with dbCAN, which detects the signature domains of each CAZyme family. The results were analyzed and visualized using a heatmap to examine the distribution of glycoside hydrolase (GH) families across different bacterial strains isolated from various food matrices. The term “presence” is used here to describe the relative abundance or occurrence of specific CAZyme families within these strains, reflecting the genetic potential for enzymatic activity, rather than direct gene expression.

Genomic analysis of *Lpb. plantarum* strains isolated from wild fermented plant- and animal-based matrices revealed a heterogeneous distribution of GH families, whose presence was associated with both genomic clustering and the matrix of origin. These enzymes, which are fundamental to the degradation of complex carbohydrates, are critical for microbial adaptation and sustaining functionality in fermented food ecosystems [67,68]. Among the identified GH families (Figure 6), GH13 and GH1 were the most prevalent, especially in strains from plant-derived substrates such as millet, kombucha, and sourdough. These enzymes are central to starch and sugar catabolism, contributing to energy release, flavor development, and metabolic flexibility required for the fermentation of complex plant matrices [69,70]. In contrast, GH families such as GH109, GH39, GH20, GH43, GH78, GH36, GH126, GH125, GH92, GH38, GH31, GH2, and GH65 appeared in more restricted genomic clusters, suggesting specialized ecological roles. Moderately represented families, namely GH23, GH42, GH32, GH73, GH25, GH6, and GH4, showed distinct distributions. GH25 was predominant in the orange cluster (10), while GH32, GH135, and GH126 were more frequent in the red and brown clusters. The purple cluster was enriched in GH78, GH43, GH20, and GH39, whereas GH109 was uniquely present in the orange cluster. These patterns highlight the ecological imprinting and functional specialization of the GH repertoire in *Lpb. plantarum*. From a technological perspective, GH13 and GH1 are particularly relevant to clean-label applications, supporting enhanced texture, sugar transformation, and the release of bioactive compounds [71]. GH1 facilitates β-glucosidase activity and the liberation of antioxidant phenolic compounds [72]. GH4 and GH43, which are enriched in plant-derived isolates, enable the utilization of phosphorylated sugars and arabinose-rich compounds [19]. In contrast, dairy-associated strains featured higher levels of GH25 and GH73, enzymes implicated in autolysis, flavor development, and survival in protein-rich, acidic environments [73,74]. GH32, which is more abundant in animal-based strains, contributes to fructan and inulin degradation, potentially enhancing prebiotic functions in dairy fermentation [75]. From a biotechnological perspective, plant-derived *Lpb: The plantarum* isolates exhibit a broader and more versatile enzymatic repertoire, which is particularly well-suited for the fermentation of complex carbohydrate-rich substrates. This functional plasticity renders them promising candidates for the development of clean-label functional foods that align with contemporary consumer demands for additive-free, plant-forward, and health-oriented products. In parallel, animal-derived strains retain niche-adapted enzymatic traits, notably those associated with cell wall hydrolases and lactate metabolism, which are indispensable in traditional dairy fermentation. Thus, the two ecological groups offer complementary enzymatic profiles: the broader GH diversity in plant-based isolates provides a strategic resource for innovation in sustainable, plant-centric food systems, whereas the specialized capabilities of dairy-associated strains support the refinement and preservation of conventional fermentation processes. This enzymatic landscape, shaped by wild fermentation environments, highlights the metabolic adaptability of *Lpb. plantarum* and its potential to transcend the functional limitations of standardized starter cultures, thereby driving the next generation of food innovation.

### 3.5. Diversity of Antimicrobial Compounds in Lactiplantibacillus plantarum Strains: Insights from WGS Analysis

Bioinformatics analysis performed using BAGEL4 allowed the assessment of potential antimicrobial compound production across various genomic clusters. The results obtained from the bioinformatic analysis were visualized and presented as a heatmap (Figure 7), which provides a comprehensive overview of antimicrobial compound production across various genomic clusters.

Bioinformatics analysis of the genomes revealed a varied and ecologically structured distribution of genes encoding antimicrobial compounds, underscoring their potential to redefine microbial functionality within food systems. Notably, the complete absence of genes encoding *Helveticin* and *LSEI_2386* across all analyzed clusters represents a deviation from the profiles previously reported in other *Lactobacillaceae* strains, suggesting that strains adapted to wild fermentation environments may rely on a distinct set of antimicrobial mechanisms. *Helveticin*, a high-molecular-weight bacteriocin with potent activity against Gram-positive bacteria, and *LSEI_2386*, a less characterized antimicrobial gene, have been associated with protective roles in dairy and gut-related contexts [76,77]. Their absence in these genomes points to niche-specific selective pressures and highlights the need for deeper exploration of antimicrobial diversity beyond conventionally studied taxa and environments. Conversely, several clusters encoded genes for a repertoire of antimicrobial peptides, including well-characterized bacteriocins with known bioactivities. In particular, the red cluster featured *Plantaricin J*, *Carnocin CP52*, *Enterocin X chain beta*, and a putative bacteriocin, forming a robust antimicrobial arsenal. *Carnocin CP52*, originally isolated from *Carnobacterium piscicola*, is particularly interesting for its stability and potential for use in refrigerated foods [78,79]. *Enterocin X*, produced by *Enterococcus faecium*, has demonstrated inhibitory effects against *Listeria monocytogenes* and other Gram-positive pathogens [80,81]. The identification of such compounds in *Lpb. plantarum* strains isolated from wild fermented matrices extend the known distribution of these genes and support co-evolution in multi-species fermentation ecosystems. *Plantaricin J* was uniquely abundant in the brown cluster, while the green and orange clusters encoded *Enterolysin A*, a murein hydrolase involved in bacterial cell lysis that is often associated with microbial competition and biofilm dynamics. The orange cluster further expressed *Plantaricin E*, and the purple cluster combined *Plantaricin J*, *E*, and *Enterocin X*, suggesting an especially diverse and potentially synergistic antimicrobial capability. These findings provide several key advances over the current state-of-the-art. First, they documented the genomic landscape of antimicrobial biosynthetic genes in *Lpb. plantarum* strains isolated from diverse wild fermentations, a context that is still underrepresented in genomic surveillance. Second, they indicated that bacteriocin gene profiles may be shaped by the type of fermentation matrix, with *plant-based strains*—especially those from sourdough and millet—showing higher richness and co-occurrence of multiple antimicrobial genes, including *Plantaricin J* and *E*. This observation reflects a possible evolutionary adaptation to environments with complex carbohydrate structures and microbial competitors, where effective niche colonization requires competitive exclusion. In contrast, animal-derived strains adapted to protein- and lipid-rich environments, like dairy products, showed a higher frequency of genes such as *Enterolysin A* and *Enterocin X*, which may play roles not only in microbial antagonism but also in autolysis-mediated flavor development and microbial succession. From a food innovation perspective, these findings open up new avenues for research. Bacteriocins, such as *Plantaricin* and *Enterocin,* are natural antimicrobials with high potential for replacing synthetic preservatives in line with clean label principles [39,82,83]. Their use supports the development of safer, minimally processed, and shelf-stable fermented products without the use of chemical additives. Furthermore, the diverse antimicrobial profiles uncovered here could be exploited in tailored starter cultures or bioprotective adjuncts, especially in plant-based products that are rapidly growing in consumer demand. In the broader context of antimicrobial resistance (AMR) and sustainable biopreservation, these naturally encoded peptides represent an ecologically grounded alternative to conventional antimicrobials. Their specificity, safety, and origin from food-grade microbes make them particularly attractive for future applications not only in food microbiology but also in clinical and nutraceutical settings. Future research should prioritize the experimental validation of these genes, including expression analysis and activity assays under food-like conditions. Moreover, exploring their regulation, co-expression patterns, and interactions with other microbial metabolic pathways could uncover synergies that are valuable for both microbial ecology and biotechnological exploitation. The results highlight a promising reservoir of antimicrobial genes in *Lpb. plantarum* strains from wild fermentations. Their diversity, ecological specificity, and alignment with current food innovation trends position them as key candidates for the ongoing transition toward safer, more natural, and sustainable food systems.

## 4. Conclusions

This study highlights the significant role of *Lpb. plantarum* strains isolated from wild fermentation environments to advance food innovation. Isolates from both plant-based matrices, such as fermented millet, kombucha, and sourdough, and animal-based matrices, including wild fermented mountain milk and natural whey starters, exhibit remarkable genetic and functional diversity. These environments, characterized by minimal technological intervention, foster microbial communities with unique adaptive traits, offering valuable resources for the food industry. Notably, genomic analysis revealed a strong expression of CAZymes, particularly from families GH13 and GH1, which are essential for metabolizing complex carbohydrates. These enzymes are crucial not only for microbial growth but also for enhancing fermentation processes and developing novel food products. Additionally, families such as GH25, GH32, and GH4 were present at varying degrees, with certain clusters exhibiting higher enzyme activity. These findings suggest that strains from wild fermentation environments possess metabolic flexibility that can be leveraged to innovate and optimize fermentation-based food production. Moreover, antimicrobial compound production genes were detected, particularly in strains from animal-based matrices, indicating their potential for biopreservation and food safety applications. This is particularly relevant in the context of antimicrobial resistance (AMR), where naturally occurring antimicrobial producers may offer sustainable alternatives to synthetic additives. Overall, this research underscores the importance of exploring natural fermentation niches as sources of microbial diversity essential for developing new and differentiated food products. This study demonstrates that wild fermentation environments harbor strains with unique functional properties that can enhance food preservation, probiotic applications, and the development of clean-label, additive-free products. This approach not only supports food innovation but also contributes to the sustainability of the food industry.

## Figures and Tables

**Figure 1 foods-14-01765-f001:**
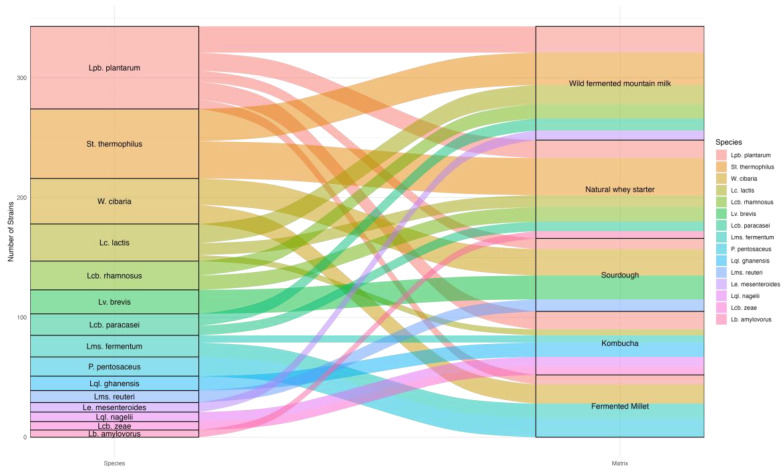
Frequency distribution of lactic acid bacteria species isolated from animal-based (wild fermented mountain milk and natural whey starter) and plant-based (sourdough, kombucha, and fermented millet) matrices. The plot illustrates the occurrence of 15 distinct lactic acid bacteria species across the five fermentation sources. The size of each box representing bacterial species is proportional to the number of isolates corresponding to that species, while the boxes denoting the sources of isolation are scaled to reflect the relative abundance of each source. The frequency of bacterial species across the different matrices highlights the variability in microbial diversity associated with plant-based and animal-based fermentation environments.

**Figure 2 foods-14-01765-f002:**
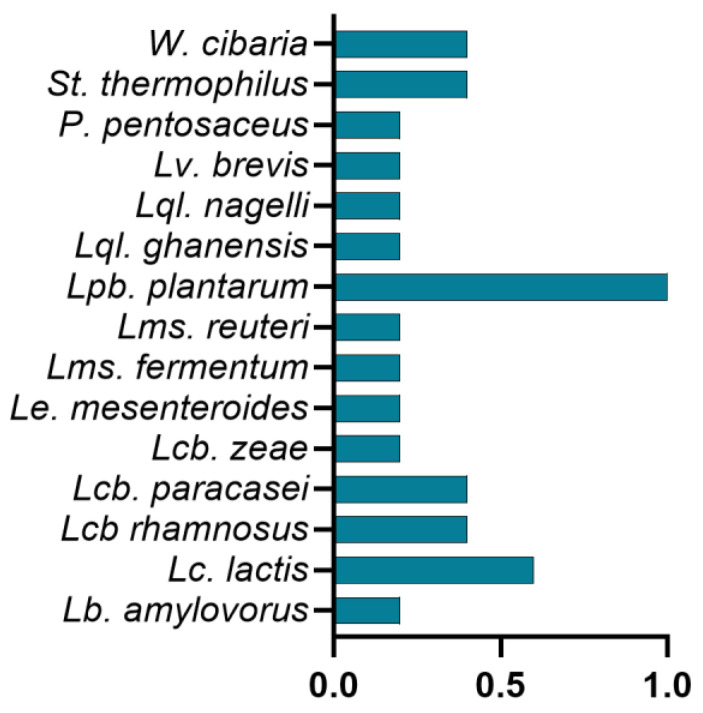
Occurrence Index (OI) of lactic acid bacteria species isolated from plant-based and animal-based sources. The OI, a dimensionless value ranging from 0.2 to 1, quantifies the relative prevalence of 15 lactic acid bacteria species isolated from three plant-based matrices (fermented millet, kombucha, and sourdough) and two animal-based matrices (wild fermented mountain milk and natural whey starters). An OI value closer to 1 indicates a higher frequency of occurrence, while values closer to 0.2 reflect a lower prevalence of the species across the analyzed matrices.

**Figure 3 foods-14-01765-f003:**
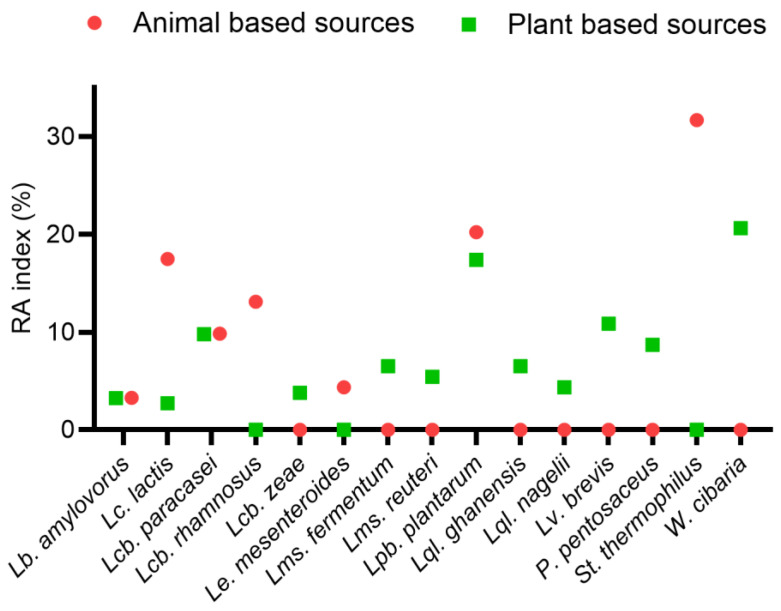
Relative abundance (RA) of 15 lactic acid bacteria species in plant-based and animal-based sources. This figure displays the relative abundance (RA) of 15 lactic acid bacteria species isolated from three plant-based matrices (fermented millet, kombucha, and sourdough) and two animal-based matrices (wild fermented mountain milk and natural whey starter). The RA illustrates the relative distribution of species, with green plots representing plant-based matrices and red plots representing animal-based matrices. Higher RA values indicate a greater prevalence of species within each matrix type, reflecting significant differences in microbial composition between plant- and animal-derived substrates.

**Figure 4 foods-14-01765-f004:**
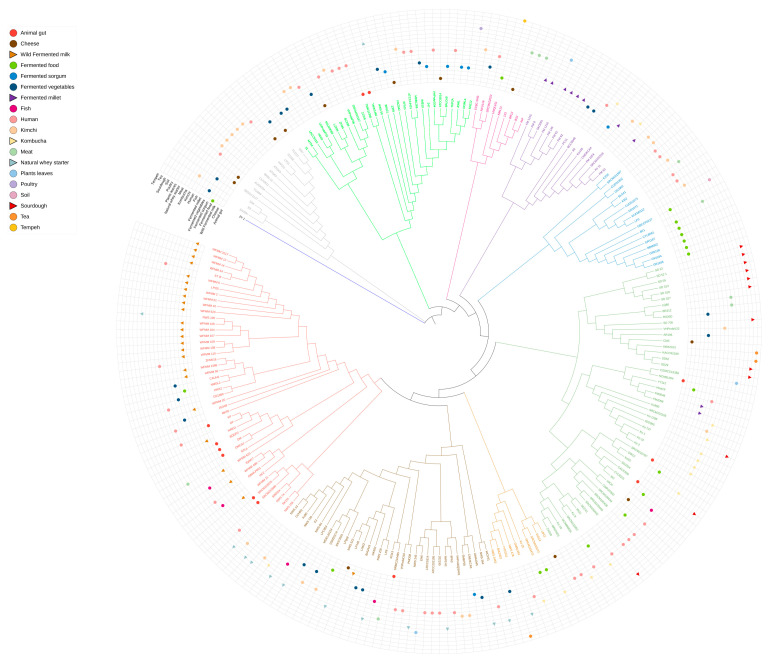
Phylogenetic tree of *Lactiplantibacillus plantarum* strains based on 233 genomes (69 newly sequenced and 164 retrieved from the NCBI database). The 69 newly sequenced strains are marked with colored triangles indicating their food matrix of origin: wild fermented mountain milk (brown), fermented millet (purple), kombucha (yellow), natural whey starter (light blue), and sourdough (red). Strains from wild fermented mountain milk and sourdough formed tight monophyletic groups, suggesting strong matrix-specific genomic adaptation. In contrast, isolates from plant-based matrices, such as kombucha, millet, and sourdough, showed broader phylogenetic dispersion than those from animal-derived matrices, highlighting the influence of ecological origin on the genetic diversity and evolutionary dynamics of *Lpb. plantarum*.

**Figure 5 foods-14-01765-f005:**
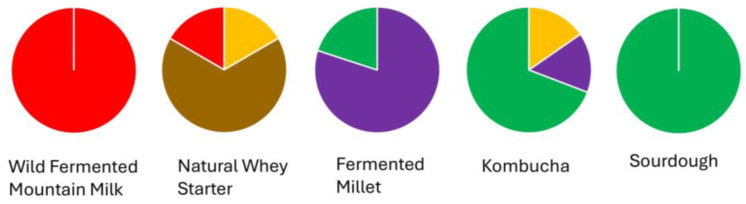
Distribution of *Lactiplantibacillus plantarum* WGS-derived sequences across phylogenetic clusters, shown as pie charts for each of the five fermentation sources—wild fermented mountain milk (*n* = 22), natural whey starter (*n* = 15), fermented millet (*n* = 10), kombucha (*n* = 15), and sourdough (*n* = 8)—with cluster affiliations color-coded as 
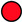
 red (dominant in milk and partly in whey), 
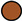
 brown (natural whey starter), 
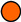
 orange (whey and kombucha), 
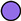
 purple (mainly millet, also kombucha), and 
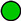
 green (sourdough, millet, kombucha), illustrating a narrower phylogenetic distribution in animal-derived matrices and higher diversity in plant-based sources.

**Figure 6 foods-14-01765-f006:**
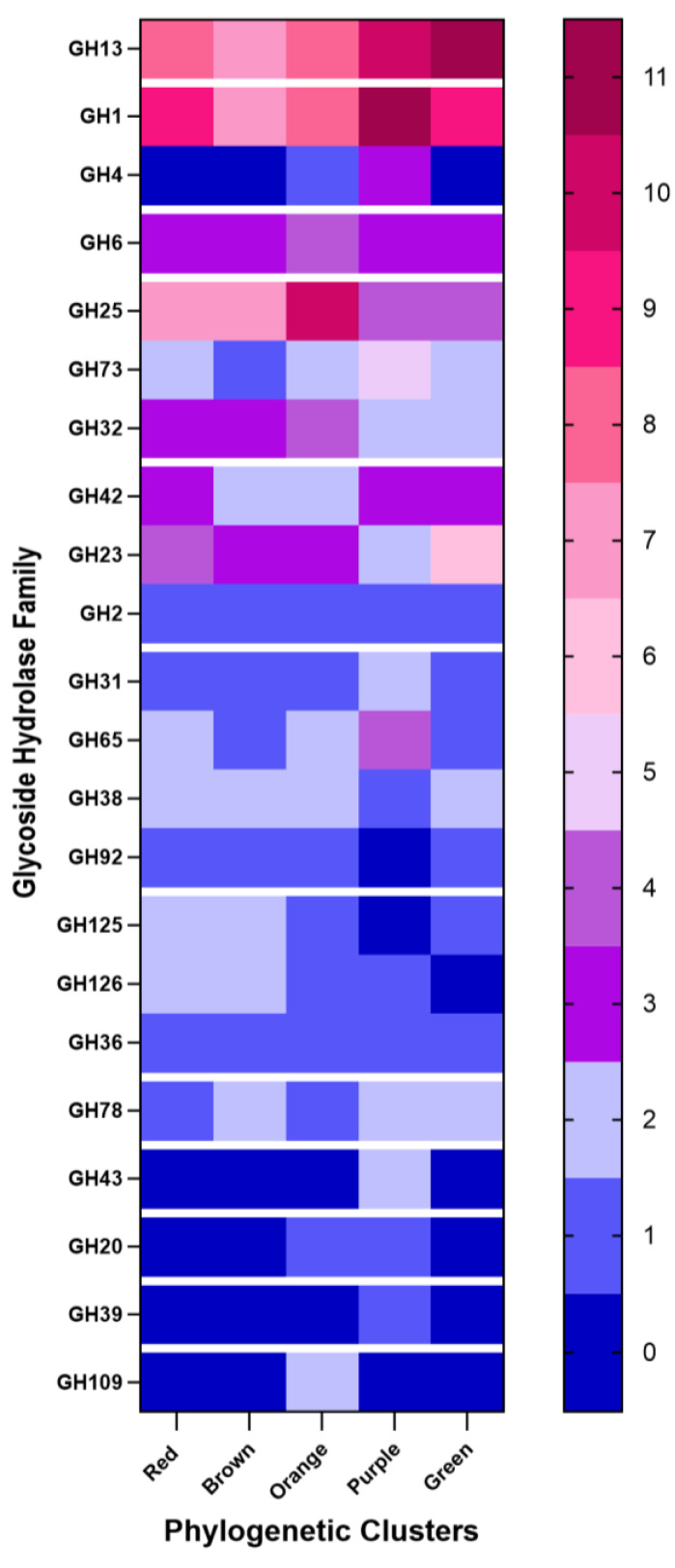
Heatmap of the glycoside hydrolase (GH) family distribution across the five *Lactiplantibacillus plantarum* WGS clusters. The map shows the relative presence of 22 GH families across five distinct *Lactiplantibacillus plantarum* WGS clusters (red, brown, orange, purple, and green). A discrete color scale is used, ranging from blue (low presence) to red (high presence), indicating the abundance of each GH family within the clusters.

**Figure 7 foods-14-01765-f007:**
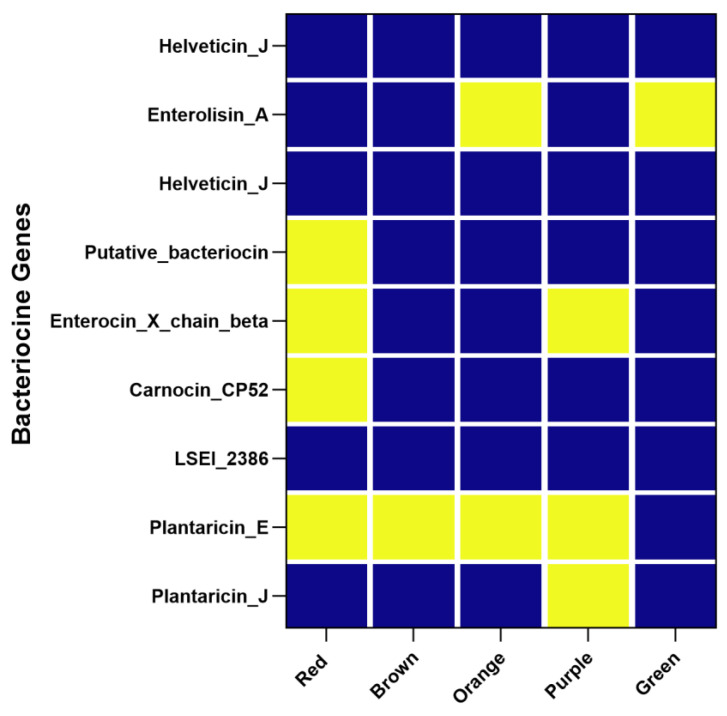
Heatmap representation of nine antimicrobial compounds or classes of antimicrobial compounds across five phylogenetic clusters (red, brown, orange, purple, and green). Yellow indicates the presence of genetic information for a specific antimicrobial substance, and blue indicates its absence in the respective cluster.

## Data Availability

The original contributions presented in this study are included in the article. Further inquiries can be directed to the corresponding author.

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
