# Peer review of "Diversity of Lactiplantibacillus plantarum in Wild Fermented Food Niches"

_foods, 2025, doi:10.3390/foods14101765_

Round 1
Reviewer 1 Report
Comments and Suggestions for Authors
The Introduction section should be revised for better clarity and content flow, with a focus on retaining only the most impactful points. It is also recommended that the Introduction be restructured into clear paragraphs according to its content.
There are an excessive number of references in the Introduction section and throughout the manuscript. Similar articles were used; some of them could be removed. In the manuscript, articles were cited consecutively. For some sections, for each sentence, three or more articles were cited. To my mind, most of them, especially self-citations, are not related to this study. In the Introduction, the relationship between the cited references from 9 to 15 and this manuscript is not clear. The manuscript should be checked. The following article is cited in three different points, each with a distinct reference number.
- Garcia-Gonzalez, N., Battista, N., Prete, R., & Corsetti, A. Health-promoting role of Lactiplantibacillus plantarum isolated from fermented foods. Microorganisms, 2021, 9(2), 349. https://doi.org/10.3390/microorganisms9020349.
The Aim of the study should be stated more concisely.
Abbreviations should be introduced in full upon first mention, followed by their abbreviated form in parentheses. Once introduced, only the abbreviation should be used without repeating the full term or the parentheses — e.g., Line 147 and Line 149.
The Materials and Methods section is well-developed, with thorough explanations provided.
The total microbial load of the fermented sources on MRS Agar, in terms of lactic acid bacteria, may be reported in Section 3.1.
Microorganism names should be written in full at first mention.
It is recommended to check the font style and use of bold text throughout the manuscript for consistency with the journal's formatting guidelines — e.g., Line 524 and Line 597.
The Conclusion appropriately summarises the key findings of the study and highlights the significance of the research.
Author Response
We thank the reviewer for his or her time and effort in evaluating our manuscript. We appreciate the constructive comments and suggestions, which proved to be extremely valuable in improving the overall quality and clarity of the article. Below, we provide detailed responses (A) to each comment raised (C), along with the corresponding changes made to the manuscript and shown in red.
C1: The Introduction section should be revised for better clarity and content flow, with a focus on retaining only the most impactful points. It is also recommended that the Introduction be restructured into clear paragraphs according to its content.
A1: We fully agree with the Reviewer’s observation. The Introduction has been thoroughly revised to enhance clarity and logical coherence. We have streamlined the content by retaining only the most relevant and impactful information, eliminating redundancy and peripheral details. Additionally, the section has been reorganized into distinct paragraphs, each addressing a specific thematic focus: (i) the importance of microbial diversity and the broader context of the study, including the limitations of industrial strains and the need for innovation; (ii) an in-depth focus on Lactiplantibacillus plantarum, its potential, current knowledge, and existing research gaps; (iii) the central thesis of the study, outlining the main objective and the rationale behind the selection of specific fermented matrices; and (iv) the gaps in current methodological approaches and the integrative framework proposed in this work.
C2. There are an excessive number of references in the Introduction section and throughout the manuscript. Similar articles were used; some of them could be removed. In the manuscript, articles were cited consecutively. For some sections, for each sentence, three or more articles were cited. To my mind, most of them, especially self-citations, are not related to this study. In the Introduction, the relationship between the cited references from 9 to 15 and this manuscript is not clear. The manuscript should be checked. The following article is cited in three different points, each with a distinct reference number.
- Garcia-Gonzalez, N., Battista, N., Prete, R., & Corsetti, A. Health-promoting role of Lactiplantibacillus plantarum isolated from fermented foods. Microorganisms, 2021, 9(2), 349. https://doi.org/10.3390/microorganisms9020349.
A2. We have carefully reviewed the entire reference list, with particular attention to the Introduction and the sections where consecutive citations appeared. Over 20 references have been removed, including a significant number of self-citations, to retain only the most relevant and directly related sources. We have also made sure to avoid redundancy, ensuring that each cited work contributes specific and non-overlapping information to the manuscript. In the Introduction section, we have reassessed the references from 9 to 15 and removed those whose connection to the central topic was not sufficiently strong or evident. Additionally, we corrected the issue of the same article being cited multiple times under different reference numbers. All citations have been standardized to maintain consistency and accuracy throughout the text. We believe these revisions have significantly improved the manuscript’s clarity and citation quality, aligning better with the scope and focus of the study.
C3. The Aim of the study should be stated more concisely.
A3. In the revised Introduction, the aim of the study has been integrated into the logical flow of the narrative rather than presented as a brief standalone statement. This approach was intended to improve clarity and coherence, allowing the rationale to emerge naturally from the discussion of current limitations in the use of Lactiplantibacillus plantarum and the value of spontaneously fermented matrices. The objective—to explore the genomic and functional diversity of Lpb. plantarum strains from five distinct substrates—is clearly stated and supported by a structured rationale. By linking the aim to both the identified scientific gap and the adopted methodological framework, we sought to highlight the relevance and novelty of our approach. We trust that this revised structure offers a clearer and more effective presentation of the study’s purpose.
C4. Abbreviations should be introduced in full upon first mention, followed by their abbreviated form in parentheses. Once introduced, only the abbreviation should be used without repeating the full term or the parentheses — e.g., Line 147 and Line 149.
A4. All abbreviations have been checked and are now introduced upon first mention and used consistently throughout the manuscript.
C5. The Materials and Methods section is well-developed, with thorough explanations provided.
A5. We thank the Reviewer for the positive feedback on the Materials and Methods section.
C6. The total microbial load of the fermented sources on MRS Agar, in terms of lactic acid bacteria, may be reported in Section 3.1.
A6. As requested, the total microbial load of the fermented sources on MRS agar, expressed as lactic acid bacteria, has been reported in Section 3.1.
C7. Microorganism names should be written in full at first mention.
A7. We thank the Reviewer for the comment. All microorganism names have been written in full at their first mention, as per the guideline.
C8. It is recommended to check the font style and use of bold text throughout the manuscript for consistency with the journal's formatting guidelines — e.g., Line 524 and Line 597.
A8. We have thoroughly checked the manuscript for font style and bold text usage and ensured consistency with the journal’s formatting guidelines throughout
C9. The Conclusion appropriately summarises the key findings of the study and highlights the significance of the research.
A9. We thank the reviewer for the positive comments on the results. We have further refined the text to improve its clarity and fluidity, while maintaining a strong emphasis on the key findings and their relevance to food innovation.
Reviewer 2 Report
Comments and Suggestions for Authors
This manuscript presents a thorough and methodologically sound investigation into the phylogenetic and functional diversity of Lactiplantibacillus plantarum strains isolated from wild fermented plant- and animal-based matrices. The integration of classical microbiological techniques with whole genome sequencing (WGS), phylogenomics, and functional annotation (e.g., CAZymes and bacteriocin gene mining) is commendable and timely. The findings are relevant for advancing our understanding of LAB biodiversity and for informing the development of next-generation starter cultures for clean-label and functional foods.
in section2.1 please split it into 2 paragraphs.
line 169, there is awkward phases such as "according to direct oral accounts from the producers", consider to change it.
There is some redundancy in the description of strain distribution across matrices, especially between the Results and Discussion sections (e.g., repeated mention of GH13/GH1 prevalence, plant-based diversity being higher). Streamlining these areas would enhance clarity.
consider to add NMDS and PCoA to visualize inter-matrix diversity.
Author Response
We sincerely thank the Reviewer for the positive evaluation of our manuscript and for highlighting the relevance and methodological rigor of our study. Below, we provide detailed responses (A) to each comment raised (C), along with the corresponding changes made to the manuscript and shown in red.
C1. in section2.1 please split it into 2 paragraphs.
A1. As suggested, Section 2.1 has been divided into two distinct paragraphs to enhance clarity and improve the overall readability of the text.
C2 Line 169, there is awkward phases such as "according to direct oral accounts from the producers", consider to change it.
A2. Thank you for your valuable comment. The sentences have been completely revised to improve clarity and precision. It now reads:
- According to documented declarations and producer testimonies, these dairies have preserved traditional endogenous fermentation practices and have never employed commercial or selected starter cultures
- In all cases, based on producer statements and supporting company documentation, fermentation was carried out without the use of commercial or selected starter cultures, thereby preserving the native microbial consortia distinctive to each ecological and cultural context.”
C3. There is some redundancy in the description of strain distribution across matrices, especially between the Results and Discussion sections (e.g., repeated mention of GH13/GH1 prevalence, plant-based diversity being higher). Streamlining these areas would enhance clarity.
A3. We carefully revised the manuscript and rationalized the overlap in content between the Results and Discussion sections. We reduced the repeated mentions of GH13/GH1 prevalence and emphasis on increased diversity in plant-based matrices, ensuring that each point is clearly presented only where it is most appropriate. Revisions are shown in red
C4. Consider to add NMDS and PCoA to visualize inter-matrix diversity.
A4. "Thank you for your suggestion to include NMDS and PCoA for visualizing inter-matrix diversity. While we acknowledge the value of these methods in exploring the relationships between matrices, we decided to focus on ecological indices such as the Shannon Index, Pielou’s Evenness, Occurrence Index, and Relative Abundance. These indices provide a comprehensive understanding of species diversity and distribution without adding complexity to the analysis. We believe this approach allows for a clearer and more fluid presentation of our findings. However, we appreciate your input, and we will consider incorporating these methods in a future work.
Reviewer 3 Report
Comments and Suggestions for Authors
Investigating the genetic and functional diversity of Lactobacillus plantarum (Lpb. plantarum) strains isolated from wild fermented foods holds significant research value for identifying traits beneficial to food innovation. However, the manuscript in its present form requires some revisions as suggested below.
Line 387-389: To quantify the ecological distribution of LAB species, we calculated the Occurrence Index (OI), defined as the proportion of matrices in which a given species was detected. Can the ecological distribution of LAB species be quantified with only 69 newly sequenced strains?
Figure 4. Phylogenetic tree of Lactiplantibacillus plantarum strains based on 233 genomes (69 newly sequenced and from the NCBI database). Strains from wild fermented mountain milk (red) and sourdough (dark green) form tight monophyletic groups. Which strains were from wild fermented mountain milk, sourdough and other fermented foods?
3.5 Diversity of Carbohydrate-Active Enzymes in Lactiplantibacillus plantarum Strains: Insights from WGS Analysis: not Carbohydrate-Active Enzymes, it should be the antimicrobial compound.
Author Response
We sincerely thank the Reviewer for the time dedicated to evaluating our manuscript, as well as for the valuable comments and positive feedback. Below, we provide our responses (A) to each specific comment (C).
C1. Line 387-389: To quantify the ecological distribution of LAB species, we calculated the Occurrence Index (OI), defined as the proportion of matrices in which a given species was detected. Can the ecological distribution of LAB species be quantified with only 69 newly sequenced strains?
A1. We acknowledge the reviewer’s observation and agree that a larger dataset would strengthen the robustness of the ecological distribution analysis. In response, we have revised the text to clarify this point as follows (Lines xxx–xxx): “To quantify the ecological distribution of LAB species (343 strains belonging to 15 different species), we calculated the Occurrence Index (OI), defined as the proportion of food matrices in which a given species was detected (Figure 2). Notably, Lpb. plantarum—represented by 69 strains—was the only species detected in all five food matrices (OI = 1.00), highlighting its remarkable ecological versatility. Although the dataset comprises only 343 newly sequenced strains, which may limit the overall comprehensiveness of the analysis, these strains were isolated from a wide array of spontaneously fermented foods, reflecting diverse ecological contexts. Therefore, while a larger dataset would further enhance the robustness of the findings, the current approach provides a valuable and informative preliminary assessment of LAB species distribution across heterogeneous fermented food environments”
C2. Figure 4. Phylogenetic tree of Lactiplantibacillus plantarum strains based on 233 genomes (69 newly sequenced and from the NCBI database). Strains from wild fermented mountain milk (red) and sourdough (dark green) form tight monophyletic groups. Which strains were from wild fermented mountain milk, sourdough and other fermented foods?
A2. We thank the reviewer for the observation. As indicated in the figure legend, the strains isolated from the five categories of fermented foods are marked in the phylogenetic tree using specific symbols and colors to distinguish them from those retrieved from the NCBI database. To clarify the origin and the individuation of the strains included in the phylogenetic analysis, the caption of Figure 4 has been revised as follows: Figure 4. Phylogenetic tree of Lactiplantibacillus plantarum strains based on 233 genomes (69 newly sequenced and 164 retrieved from the NCBI database). The 69 newly sequenced strains are marked with colored triangles indicating their food matrix of origin: wild fermented mountain milk (brown), fermented millet (purple), kombucha (yellow), natural whey starter (light blue), and sourdough (red). Strains from wild fermented mountain milk and sourdough form tight monophyletic groups, suggesting strong matrix-specific genomic adaptation. In contrast, isolates from plant-based matrices such as kombucha, millet, and sourdough show broader phylogenetic dispersion compared to those from animal-derived matrices, highlighting the influence of ecological origin on the genetic diversity and evolutionary dynamics of Lpb. plantarum
C3. 3.5 Diversity of Carbohydrate-Active Enzymes in Lactiplantibacillus plantarum Strains: Insights from WGS Analysis: not Carbohydrate-Active Enzymes, it should be the antimicrobial compound.
A3. We thank the reviewer for pointing this out. We agree that the section title should reflect the actual content, which focuses on antimicrobial compounds rather than carbohydrate-active enzymes. Accordingly, the title of Section 3.5 has been revised to: “3.5 Diversity of Antimicrobial Compounds in Lactiplantibacillus plantarum Strains: Insights from WGS Analysis.” The correction has been implemented in the revised manuscript.